# Effect of Template–Mediated Alumina Nanoparticle Morphology on Sapphire Wafer Production via Heat Exchange Method

**DOI:** 10.3390/ma16175938

**Published:** 2023-08-30

**Authors:** Yadian Xie, Miaoxuan Xue, Lanxing Gao, Yanqing Hou, Bo Yang, Xin Tong

**Affiliations:** 1Guizhou Provincial Key Laboratory in High Education Institutions of Low-Dimensional Materials and Environmental and Ecological Governance, Key Laboratory of Low-Dimensional Materials and Big Data, College of Chemical Engineering, Guizhou Minzu University, Guiyang 550025, China; xuemx0418@163.com (M.X.); glxprettylife@163.com (L.G.); hhouyanqing@163.com (Y.H.); 2Faculty of Metallurgy and Energy Engineering, Kunming University of Science and Technology, Kunming 650093, China; 3School of Chemistry and Material Science, Guizhou Normal University, Guiyang 550014, China

**Keywords:** spherical alumina, templating synthesis, heat exchange method, sapphire crystal, GaN–based LED

## Abstract

The sapphire crystal, the most commonly used LED substrate material, has excellent optical and chemical properties and has rapidly developed in recent years. However, the challenge of growing large–size sapphire crystals remains. This paper presents a novel approach using alumina nanoparticles synthesized with abietic acid as a template to enhance sapphire growth via the heat exchange method. This study explores the effects of temperature, time, and template amount on the structure and morphology of the synthesized alumina nanoparticles. The results show that the morphology of the raw material, particularly spherical alumina nanoparticles, positively affects the quality and yield stability of sapphire products. Furthermore, the light output power of GaN–based LED chips made with the experimentally fabricated sapphire substrate increased from 3.47 W/µm^2^ to 3.71 W/µm^2^, a 6.9% increase compared to commercially available sapphire substrates. This research highlights the potential of using abietic acid as a template for alumina nanoparticle synthesis and their application in sapphire growth for LED production.

## 1. Introduction

As the most commonly used LED substrate material, the sapphire crystal has been rapidly developed in recent years because of its excellent optical properties, mechanical properties, and other properties. There are many methods to produce sapphire crystals in industry, such as the Czochralski (Cz), Kyropoulos (Ky), edge-defined film-fed growth (EFG) [1], chemical vapor deposition (CVD) [2,3] and heat exchanger methods (HEM) [4]. The heat exchange method is a crystal growth technique for large-size sapphires. The principle is to use the heat exchanger to remove heat, so there will be a longitudinal temperature gradient that is cold in the lower section and warm in the upper section in the crystal growth area [5]. Then, the temperature gradient is controlled by controlling the rate of the gas flow in the heat exchanger (He cooling source) and heating power level. In this way, the melt in the crucible will be slowly solidified into a crystal in a down–up direction. The heat exchange method has high requirements for the quality of raw materials in order to form a very stable temperature field in the crystal growth process and the segregation coefficient of the impurity in the melt is controlled to be less than 1. Therefore, the impurities will continuously be discharged into the melt from the solid–liquid interface and finally distributed in the outer layer of the surface where the crystal and the crucible wall are in contact with each other.

In 1970, Schmid and Viechnicki [6] first grew bulk sapphire crystals using the heat exchange method. Later, many researchers studied the temperature distribution, melt flow, and shape of the melt–crystal interface in the process of heat exchange solidification. Lu et al. [7] used the finite element software FIDAP to study the temperature, velocity distribution, and interface shape during the growth of large sapphire crystals prepared via the heat exchange method. It was found that the contact angle is objective before the solid–melt interface touches the sidewall of the crime. Therefore, hot spots always appear in this process The maximum convexity decreases significantly when the cooling zone radius (RC) increases. Wu et al. [1] conducted a numerical study on the shape of the melt–crystal interface during the early solidification process when the crystal diameter increased. They found that the obtuse contact angle was caused by the difference in thermal resistance between sapphire crystals and the melt, as well as the insufficient cooling effect at the bottom of the crucible, and proposed solutions.

In recent years, the research of alumina raw materials has mainly focused on the product purity [8], synthesis method [9], sintering [10], etc. Aluminum’s application on sapphire growth depends not only on the purity of the alumina powder but also on the size and shape of the particles. The common aluminum is rod-like [11], fibrous [12], flake-like [13], spherical [14], and so on.

Abietic acid (AA) is a natural tricyclic diterpenoid containing oxygen compounds. Its long-chain structure effectively prevents particle aggregation, making it a useful emulsifier, thickener, reductant, or stabilizer in food, bioproducts, and pharmaceutical formulations to control texture, microstructure, viscosity, and stability [15,16]. Sifontes Á B et al. [17] used boehmite and carboxylic acid to produce aluminocarboxane nanoparticles from rosin extract; dry carboxylic acid aluminum oxide nanoparticles were fired at a temperature of 650 °C.

In this study, spherical alumina nanoscale particles were successfully prepared using AlOOH as a precursor, (NH_2_)_2_CO as the precipitant, and AA as a template by adjusting the AA/urea weight ratio. The effect of the synthesis temperature, time, and amount of template on the morphology of the product was also investigated. This study further discusses the effect of different morphologies of alumina as raw materials on the amount of sapphire wafers produced via the heat exchange method. Finally, the light output power of GaN–based LED chips made with the experimentally fabricated sapphire substrate increased from 3.47 W/µm^2^ to 3.71 W/µm^2^, a 6.9% increase compared to commercially available sapphire substrates

## 2. Experimental Part

### 2.1. Synthesis of Spherical Alumina

The synthesis process of alumina nanoparticles is described as follows: Initially, 0.4 mol/L aluminum nitrate solution is prepared by dissolving aluminum nitrate in ultrapure water (resistivity = 18 MΩ·cm). Abietic acid (AA) is dissolved in isopropanol, and urea is added while maintaining the AA/urea weight ratio at 1.25. The well-prepared additive solution is then added to the aluminum sulfate solution, and the resulting solution is vigorously stirred in an autoclave at 160 °C for 30 min. After that, the solution is slowly stirred for 3 h and rapidly cooled to room temperature. The obtained precursor alumina is cleaned with ultrapure water 3–5 times and dried for 1 h at 80 °C. The white spherical alumina powder obtained after calcination at 600 °C for 2 h is called intermediate Al_2_O_3_. Finally, the prepared spherical alumina powder is calcined at 1600 °C for 1 h to produce the raw material Al_2_O_3_ used for sapphire crystal growth.

### 2.2. The Growth of Sapphire Wafer

In crystal growth, the raw material powder is pressed into bulk to prevent powder loss and increase density. The furnace is cleaned and the air flow rate increased before adding raw materials. Seeds are attached to the bottom of the crucible, which is cleaned with ethanol and acetone, and placed on the heat exchanger. Bulk materials are added to the crucible one by one, and the furnace is closed for growth. Crystal growth via the heat exchange method involves heating and melting, crystal growth, and annealing and cooling. Sapphire crystal samples are processed through several steps to obtain a 2–inch sapphire wafer. In the experiment, intermediate Al_2_O_3_ with different morphologies is used to prepare sapphire crystal Al_2_O_3_, and the influence on sapphire growth via the heat exchange method is studied by measuring the average length of the crystal rods produced by each intermediate Al_2_O_3_.

### 2.3. Preparation of GaN-Based LED Device

GaN-based LED epitaxial wafers are grown using metal organic vapor deposition (MOCVD) equipment. The growth process uses H_2_ and N_2_ as carrier gases. Trimethylgallium (TMGa), trimethylindium (TMIn), and ammonia (NH_3_) are the Ga source, In source, and N source, respectively. Magnesocene (CP_2_Mg) and silane (SiH_4_) are dopant materials for p–type gallium nitride (p–GaN) and n–type gallium nitride (n–GaN), respectively. ITO thin films are deposited on the mold capping layer via electron beam evaporation. The raw material for evaporation is bulk ITO, and the mass ratio of In_2_O_3_ and Sn_2_O_3_ is 9:1. The Ti/Al/Ni/Au and Ni/Au electrodes for n-type and p–type contacts were prepared via photolithography and electron beam evaporation.

### 2.4. The Physical Characterization

X-ray diffraction (XRD, PANalytical X’Pert PRO, Almelo, Holland) and a scanning electron microscope (SEM, HITACHI SU8600) were used to study the crystallinity and morphology of prepared powders and sapphire chips. Transmission electron microscopy (TEM) measurements were carried out on a Tecnai G2 F20 (FEI, Hillsboro, OR, USA) operated at 200 kV. All samples were first dispersed in ethanol and then collected using a copper grid covered with carbon films for measurements. A transmittivity test apparatus (UV–VIS–NIR) is a test method for characterizing the optical transmittance of crystals in the UV, visible, and near-infrared bands, which is the main characterization method for the optical properties of crystal. The main principle of the test is the Beer–Lambert Law; that is, after a certain band of electromagnetic waves is transmitted through the crystal materials, the electromagnetic waves’ remaining intensity after the absorption, scattering, reflection, and the like are deducted is calculated. The transmittance of sapphire wafers at room temperature within 190~1100 nm was measured using the Cary60 type UV–VIS–NIR (Agilent, Santa Clara, CA, USA) transmittance tester. The optical properties of the samples were characterized via 325 nm laser photoluminescence (PL) spectroscopy (EdinburghInstruments, Livingston, UK) at room temperature. The GAMMA Scientific GS–1190 RadoMA-Lite KEITHLEY 2400 (Keithley, Cleveland, OH, USA)system was used to evaluate the electroluminescent (EL) spectroscopy of the LED.

Using AlOOH as the precursor, (NH_2_)_2_CO as the precipitant, and AA as the template, the AA/urea weight ratio was adjusted to prepare spherical alumina nanoparticles. Sapphire crystals were prepared via the heat exchange method after the precursor alumina was calcined.

## 3. Results

As shown in Figure 1, Spherical alumina nanoparticles were synthesized using AlOOH as the precursor, (NH_2_)_2_CO as the precipitant, and AA as the template, with the AA/urea weight ratio adjusted. After the calcination of the precursor alumina, sapphire crystals were grown using the heat exchange method. The reaction temperatures are 100 °C, 130 °C, 160 °C, and 190 °C and the reaction times are 3 h, 6 h, and 9 h. The AA/urea weight ratio is within the range of 0 to 2.5. The synthesis parameters for alumina are presented in Table 1 and Table 2.

XRD images of the precursor alumina under different synthesis times and temperatures are presented in Figure 2a, revealing that amorphous products A and B are obtained when the synthesis temperature is lower, such as 100 °C and 130 °C. However, precursors C and D start to crystallize and produce AlOOH (JCPDS 01–072–0359) when the temperature is raised to 160 °C and 190 °C. The results indicate that the hydrothermal system lacks sufficient energy to promote migration and crystallization at lower temperatures, resulting in the samples being amorphous. As the hydrothermal temperature increases, various diffraction peaks emerge and the intensity increases and the width narrows, indicating that the crystallinity of the particles becomes stronger. Precursors E and F, synthesized at 130 °C for longer times of 6 h and 9 h, also exhibit enough energy to crystallize. Compared with the higher temperature precursors C and D, the diffraction peaks of precursors E and F did not change much, indicating that the long reaction time can also prepare the precursor AlOOH with good crystallinity. The morphology and crystal structure of the products is primarily controlled by the average growth rate of the crystal.

Figure 2d shows the SEM images of the precursor alumina sample at 160 °C and 3 h. The sample particles are spherical and uniform in size, with smooth surfaces and particle sizes ranging from 100–300 nm. As shown in Appendix A, the reaction temperature and time have a great influence on the morphology of the samples. When the temperature was 100 °C and 130 °C, the sample particles obtained were mainly flaky, spherical, and massive (Appendix A); when the temperature reached 160 °C, the shape of the sample particles was regular and spherical with a smooth surface and the particle size was between 100 and 300 nm (Appendix A). With the increasing temperature, AA tends to carbonize when synthesized at high temperature and then loses part of its role as a template, which is not conducive to the development and formation of spherical particles. As a result, the sample becomes an irregular flocculent mass that aggregates a large number of particles and crystallizes into irregular crystals (Appendix A). Prolonged synthesis is also detrimental to the development of spherical alumina particles (Appendix A). It tends to form flocculent, streaky, and irregular particles, which affects the normal development and growth of spherical particles.

Figure 2b shows the intermediate alumina XRD after calcination at 600 °C. Samples A and B, transformed from the amorphous precursor, are both γ–Al_2_O_3_. Similarly, samples C, D, E, and F are also γ–Al_2_O_3_ (JCDPS 10–0425). The diffraction peak position does not change with the increase in hydrothermal temperature, but the intensity increases and the width narrows. This suggests that the amorphous precursor prepared at a lower synthesis temperature requires more energy to promote grain crystallization and transform into γ–Al_2_O_3_. Figure 2e shows the SEM image of intermediate alumina, from which it can be seen that the spherical alumina impurities have been removed after sintering at 600 °C, and the surface becomes relatively smooth and clean.

Figure 2c shows the XRD image of sapphire crystal raw material alumina after calcination at a high temperature of 1600 °C, indicating that the α–alumina has a single crystal phase, highly ordered crystal growth, and good crystallinity. All samples are transformed into α-phase alumina (JCPDS10–0173). Figure 2f shows the TEM image of pristine alumina of sapphire crystals after calcination at 1600 °C. The particle size ranges from 100 to 300 nm, indicating that the size of the spherical alumina remains unchanged after high-temperature calcination. The research on synthesis temperature and time indicates that the intermediate alumina prepared at 130 °C for 3 h and calcined at 600 °C has the best spherical shape, with a particle size mainly within 100–300 nm. The α-alumina obtained after calcination at 1600 °C is the raw material for sapphire growth via the heat exchange method. The performance and yield of the sapphire wafer will be tested.

In the process of synthesis, the quality of the template has a very important influence on the morphology of the samples. Table 3 shows the test results of different weight ratios in the preparation of spherical alumina (AA/urea).

Table 3 and Appendix A show that when the AA/urea weight ratio is 0 the intermediate Al_2_O_3_ is stripped and irregularly rectangular with random distribution and accumulation after calcination at 600 °C. The particle sizes are between 1 μm and 2 μm. The density of raw material Al_2_O_3_ is 3.52 g/cm^3^ after continuously calcinating at 1600 °C. With the gradual increase in the AA/urea weight ratio, the intermediate Al_2_O_3_ gradually becomes spherical, the particle size decreases, and the density of raw material Al_2_O_3_ gradually increases. Finally, when the AA/urea weight ratio becomes 1.25, the intermediate Al_2_O_3_ become regular spherical, the particles sizes are within the range of 50 nm to 300 nm, and the density of the raw material Al_2_O_3_ has increased to 3.85 g/cm^3^. However, if the weight ratio continues to increase, the excess amount of AA will cause a high solution viscosity in the synthesis, resulting in a weakened template steric effect and reduced homogeneous nucleation. In this case, the particles are not conducive to dispersing. The experiment results show that the particle size and shape of the intermediate have a direct impact on the density of raw material Al_2_O_3_. It shows that after high-temperature calcination the spherical intermediate Al_2_O_3_ with a particle size between 50 nm and 300 nm can significantly increase the density of raw material Al_2_O_3_.

In the experiment, raw material Al_2_O_3_ produced by intermediate Al_2_O_3_ with the same purity and different morphologies after calcination at 1600 °C was used as the raw material of sapphire growth via the heat exchange method. The effects of the raw material morphology and density on the total length, mechanical properties, and optical properties of the sapphire rod were studied.

Figure 3a shows the stacking density of intermediate Al_2_O_3_ with different morphology and the total length of the prepared sapphire rod. It can be seen from the figure that the packing densities of rod–like, partially spherical, partially massive, and partially spherical intermediates are significantly smaller than those of spheroids, and the resulting rods are relatively short in length. The average length is about 3200 mm to 3800 mm. As the figure shows, there is a large standard deviation in the length of the crystal rod, which means that the length of the crystal rod has a great fluctuation. The crystal rods produced by these three types of intermediates are short, and the length range is not stable, while the crystal rod produced by the spherical intermediate shows that the length of the crystal rod is relatively stable, remaining around 4000 mm. From the experimental results, we find that when the purity of raw materials is the same the morphology of the raw material has a great influence on the total length and the length stability of the sapphire crystal rod produced via the heat exchange method. Compared to rod-like and spherical or massive and spherical raw materials, spherical raw materials can produce longer sapphire crystals, and the length is more stable. The experiment tested the sapphire crystal rods and wafers.

To evaluate the mechanical properties of sapphire rods, the yield strength and elastic modulus of sapphire rods prepared from intermediate Al_2_O_3_ with different morphologies are measured (Figure 3b). In general, the crystal rod produced by the spherical intermediate demonstrates extraordinarily greater mechanical strength than other shapes of intermediate Al_2_O_3_.

According to Figure 3c, the XRD test results of the c–plane (0001) sapphire crystal rod show that it has reached a very steep diffraction peak value when the 2 theta angle is 41.6623°, which indicates that the crystal rod has a complete sapphire crystal structure. Figure 3d is the transmission spectrum of the sapphire c–plane (0001) wafer. The transmittance of the wafer through 300–400 nm, the range of ultraviolet, is more than 80%, and through 500–1000 nm it is more than 85%. This indicates that using the spherical alumina nanoparticles prepared using an AA template as the raw material of sapphire growth via the heat exchange method can produce sapphire crystal products of good quality and a stable length of the sapphire crystal rod.

The PL spectra of GaN–based LED chips were collected at room temperature. As shown in Figure 4a, the GaN–based LED chip prepared on the sapphire substrate for this experiment observed sharp and strong band-edge emission at 445 nm with an FWHM of 26.0 nm, while the GaN–based LED chip prepared on the commercially available sapphire substrate observed a wider and weaker band-edge emission at 449 nm with an FWHM of 33.6 nm, because the InGaN/GaN MQW of both samples is similar. The red shift observed in the commercially available GaN–based LED chips prepared on sapphire substrates may be caused by the quantum confinement Stark effect (QCSE) induced by the piezoelectric field in the strained InGaN/GaN MQW [18]. It is shown that high-quality GaN–based LED chips were achieved on sapphire substrates using the MOCVD method, and the GaN–based LED chips prepared on sapphire substrates in this experiment exhibit superior PL luminescence than those prepared on commercially available sapphire substrates. EL measurements of the GaN–based LED chips were also performed at room temperature. Figure 4b shows the EL spectra at an injection current of 20 mA, and it can be observed that the GaN–based LED chip prepared on this experimental sapphire substrate has a strong band–edge emission at 449 nm with an FWHM value of 23.0 nm, while the commercially available GaN–based LED chip prepared on sapphire substrate has a weaker band–edge emission at 449 nm with a half-height width of 23.3 nm. The high EL emission intensity confirms the GaN–based LED has good hole injection, mainly due to the high NA–ND value in Mg-doped p–GaN thanks to our proposed device structure [19,20,21]. Furthermore, the high electroluminescence intensity confirms the good electroluminescence of the GaN–based LED chips prepared from the sapphire substrate prepared in this experiment. The epitaxial wafer samples were made into LED chips according to the conventional process, and then the variation in the light output power (L_OP_) and voltage (V) with the current density (J) was tested. As shown in Figure 4c,d, it can be found that the light output power of both samples increases with increases in the injection current. For the LED chip on the sapphire substrate prepared in this experiment, the light output power reaches a maximum of 3.71 W/µm^2^ at a current density of 3.5 A/mm^−2^, while for the LED chip on a commercially available sapphire substrate the light output power reaches a maximum of 3.71 W/µm^2^ at a current density of 4 A/mm^−2^, and the turn-on voltage of both samples is basically equivalent at 2.4 V.

## 4. Conclusions

In this study, 50–300 nm spherical nanoscale alumina particles were prepared with good dispersion and uniform morphology using a template method with a 1.25 AA/urea weight ratio. We have noted that the precursor prepared under a lower temperature of 130 °C and for a shorter time of 3 h in the synthesis has a better morphology to form spherical particles. When this type of precursor calcines at 600 °C, the inorganic Al^3+^ is more likely to interact and connect with the organic micelle interface through electrostatic interaction, covering the entire surface of aluminum ions, so that all the surfaces have the same growth speed, the homogeneous nucleation occurs, and spherical particles are prepared. At the same time, the experimental results show that the different morphologies of the raw materials have an important influence on the yield and quality of the sapphire crystal rod. Spherical alumina nanoparticles prepared using this method can produce sapphire crystal rods with total lengths between 4000 mm and 4500 mm and a higher transmittance of 85%. Furthermore, the light output power of the GaN–based LED chip made of the sapphire substrate is as high as 3.71 W/µm^2^. The method and the research results have not only described a new template method to control the morphology of the sapphire crystal materials but, more importantly, they have also provided a new idea for the research of raw materials in the field of sapphire crystal growth.

## Figures and Tables

**Figure 1 materials-16-05938-f001:**
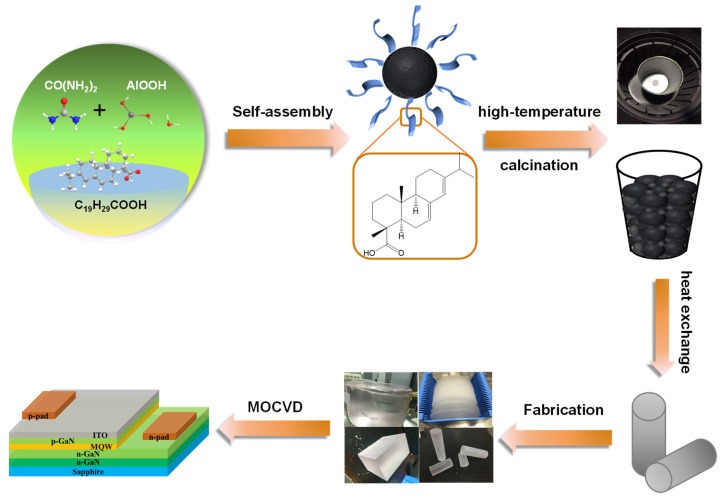
Schematic diagram of the sapphire synthesis process and LED device.

**Figure 2 materials-16-05938-f002:**
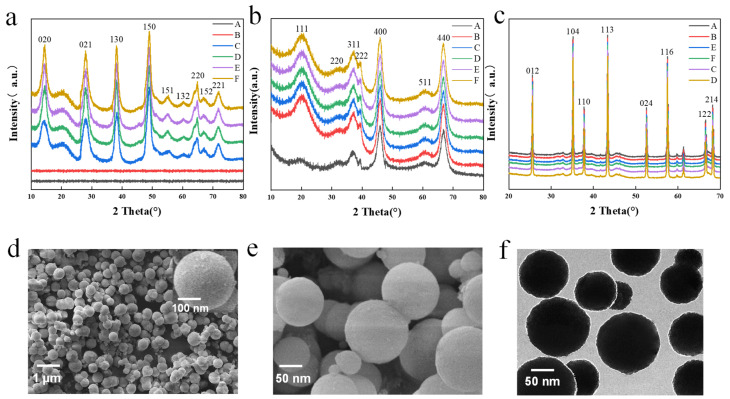
(**a**) Precursor alumina XRD image under different synthesis conditions: (A) Amorphous–100–3; (B) Amorphous–130–3; (C) AlOOH–160–3; (D) AlOOH–190–3; (E) AlOOH–130–6; (F) AlOOH–130–9. (**b**) XRD image of intermediate alumina of precursor alumina at 600 °C under different synthesis conditions: (A) Al_2_O_3_–100–3; (B) Al_2_O_3_–130–3; (C) Al_2_O_3_–160–3; (D) Al_2_O_3_–190–3; (E) Al_2_O_3_–130–6; (F) Al_2_O_3_–130–9. (**c**) Sapphire crystal raw material alumina XRD image under different synthesis conditions in the calcination of 1600 °C: (A) Al_2_O_3_–100–3; (B) Al_2_O_3_–130–3; (C) Al_2_O_3_–160–3; (D) Al_2_O_3_–190–3; (E) Al_2_O_3_–130–6; (F) Al_2_O_3_–130–9. (**d**) Precursor SEM image at 130 °C for 3 h. (**e**) Intermediate alumina SEM image at 600 °C. (**f**) Sapphire crystal raw material alumina TEM image at 1600 °C.

**Figure 3 materials-16-05938-f003:**
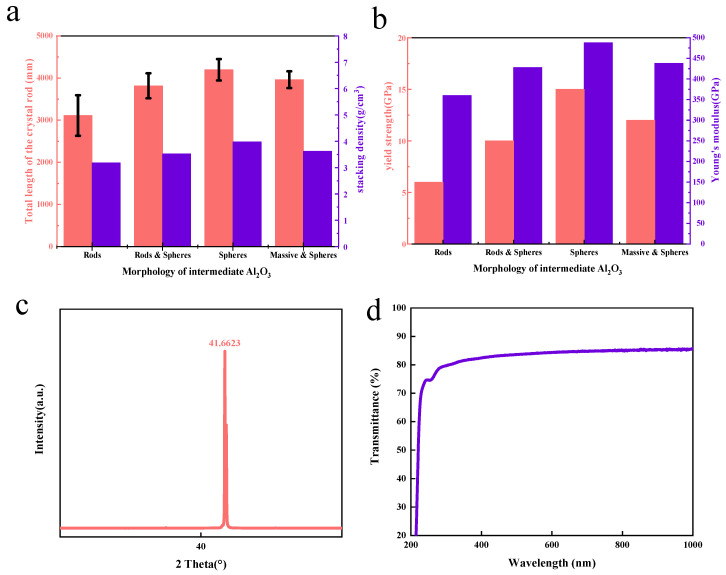
Quality of sapphire grown via heat exchange method: (**a**) stacking density of intermediate Al_2_O_3_ with different morphology and the total length of the prepared sapphire rod; (**b**) yield strength and elastic modulus of sapphire rods prepared from intermediate Al_2_O_3_ with different morphologies; (**c**) sapphire c–plane (0001) crystal rod XRD pattern; (**d**) sapphire c–plane (0001) wafer transmission spectrum.

**Figure 4 materials-16-05938-f004:**
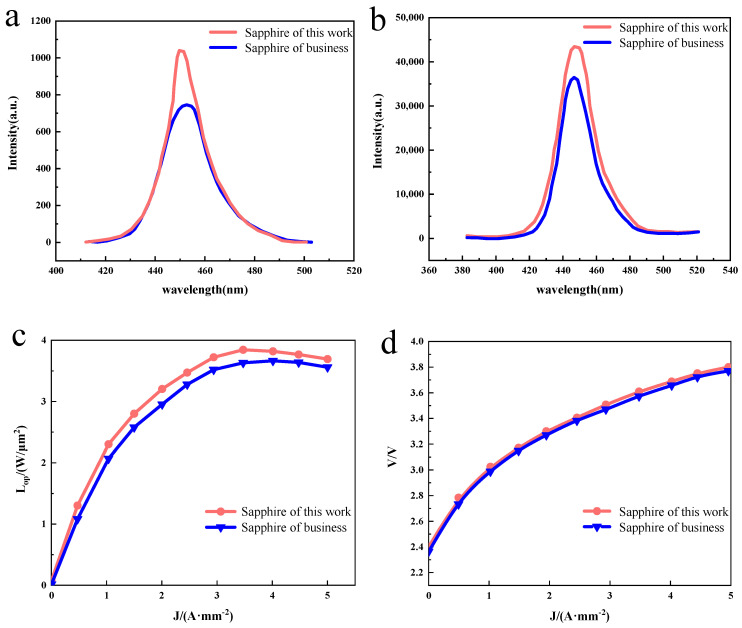
(**a**) Room–temperature photoluminescence spectroscopy of the GaN–based LED chip. (**b**) Room–temperature electroluminescent spectra of the GaN–based LED chip under the injection current of 20 mA. (**c**,**d**) Plot of optical output power and voltage as a function of current density.

**Table 1 materials-16-05938-t001:** Process parameters of alumina prepared using different reaction temperatures and reaction times.

Sample Number	Weight Ratio (AA/Urea)	Temperature (°C)	Time (h)
A	1.25	100	3
B	1.25	130	3
C	1.25	160	3
D	1.25	190	3
E	1.25	130	6
F	1.25	130	9

**Table 2 materials-16-05938-t002:** Process parameters of spherical alumina prepared using different amounts of abietic acid.

Sample Number	Weight Ratio (AA/Urea)	Temperature (°C)	Time (h)
G	0	160	3
H	0.625	160	3
I	1.25	160	3
J	2.5	160	3

**Table 3 materials-16-05938-t003:** Morphology properties of alumina with different AA/urea weight ratios.

Sample Number	Weight Ratio (AA/Urea)	Intermediate Al_2_O_3_ Particle Size (nm)	Intermediate Al_2_O_3_ Shape	Density of Crystal Growth Raw Material Al_2_O_3_ (g/cm^3^)
G	0	1500~2000	Strip, irregular rectangular	3.52
H	0.625	500~1000	Strip, partially spherical	3.66
I	1.25	50~300	Spherical	3.85
J	2.5	>500	Irregular massive, partially spherical	3.81

## Data Availability

Data sharing not applicable. No new data were created or analyzed in this study. Data sharing is not applicable to this article.

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
