# Peer review of "Effect of Template–Mediated Alumina Nanoparticle Morphology on Sapphire Wafer Production via Heat Exchange Method"

_materials, 2023, doi:10.3390/ma16175938_

Round 1

Reviewer 1 Report

Title: Effect of Template-Mediated Alumina Nanoparticle Morphology on Sapphire Wafer Production via Heat Exchange Method

Keywords must be improved: „alumina” is already in the title  „spherical shape” is too much general; „heat exchange method” is too general, also

Line 21: „…demonstrates a high light output power of 928 mW”. The units of measure must be improved, something is missing: if it is light output, lumen must be used. If it is electrical power, 928 mW must be associated with something specific (mW/cm2, for example).

Line 28: Typo: ”…. rapid development in recent years.[1-4] ”

Better: ”…. rapid development in recent years [1-4].” In all te paper….

 Citation is not correct. From line 27 to line 53 you have exhausted all 22 bibliographic references. An average of one reference per line. And after line 53 you have not used references. Citations must be used as a justification for your work, but it represent also a scientific measurement tool. Critical assessment must be used, also. Introduction must be rewritten.

Line 103: “…characterizing the optical transmittance of crystals in the UV visible near infrared band”. Better: “…characterizing the optical transmittance of crystals in the UV, visible and near infrared bands”.

Line 105: “…after a certain band of light“. You do not have only light, you have radiation (for UV and infrared). 

Line 109:”… in the range of  400~4000 cm-1”. Maybe a typo?

Figure 2 has only tree images, not six images as you explain in lines 132 to 160 (Fiure 2 a, b, c)

Figure 2d is missing (d), maybe a typo.

In Table 3 you have density (cm3/g). Better is “specific volume”, inverse of density. And in paper, also.

Line 211: ” …Figure 8 shows the functional relation between…” Figure 8 maybe in other paper….

Line 219: you must use nm instead mm!

Line 223: 4000 mm again! And in Figure 3. And in conclusions….

Reviewer 2 Report

This paper presents a novel approach using alumina nanoparticles synthesized with abietic acid as a template to enhance sapphire growth by heat exchange method. The study explores the effects of temperature, time, and template amount on the structure and morphology of the synthesized alumina nano-particles. The results show that the morphology of the raw material, particularly spherical alumina nano-particles, positively affects the quality and yield stability of sapphire products. Furthermore, the GaN-based LED chip made from sapphire substrate demonstrates a high light output power of 928 mW. This research highlights the potential of using abietic acid as a template for alumina nano-particle synthesis and their application in sapphire growth for LED production.

In my opinion, the work should be supplemented with some fundamental works devoted to sapphire. For example,

V.N. Kurlov, Sapphire: properties, growth, and applications, Reference Module in Materials Science and Materials Engineering, (2016), https://doi.org/10.1016/

In addition, it should be noted that, sapphire can also be synthesized from the vapor phase such as CVD (Chemical Vapor Deposition) or sputtering method at low temperature, but in this case the thickness of thin film obtained may be less than several μm level:

2. H.J. Jang, J.S. An, C.Y. Park, J.H. Lee, B.H. Choi, C. Hun Lee, Growth behavior of high density Al2O3 layer prepared by using cyclic chemical vapor deposition technology, J. Nanosci. Nanotechnol. 15 (2015) 5232–5237.

3. M.D. Groner, F.H. Fabreguette, J.W. Elam, S.M. George, Low-temperature Al2O3 atomic layer deposition, Chem. Mater. 16 (2004) 639–645.

In addition, it is necessary to improve the quality of some drawings, in particular, these are Figs 2, 3,4

In particular, it is necessary to increase the font and readability of the symbols.

After elimination of comments, the work can be accepted for publication.

Reviewer 3 Report

The work is not well structured.

The introduction part does not reflect the state of the art

the experimental methodology is not well described such as nano particles, growth and LED device. Moreover, no relation between them was well discussed.

English must be improve and checked by a native speaker.

So, due to serious faults in writing and structure, I can not recommend this work for publication.

Finally the device effect is not presented in the deeply way which is the most important of the work.

English must be improve and checked by a native speaker.

Reviewer 4 Report

Minor corrections are required

Round 2

Reviewer 1 Report

    The paper has obviously progressed. Almost all observations were accepted by the authors, which is a confirmation for the reviewer himself. It is an example of scientific dialogue that is much more consistent compared to the dialogue possible at specialized conferences, where the papers are presented in public. This detail is also a plus for the Materials Journal.

    Not all observations were accepted, the use of the unit of measure W for the specific emission of light is debatable and attackable. But a work can also have controversial details that are the basis of subsequent citations.

Many days have passed since the initial submission of the work, in April 2023. Even if the reviewers worked quickly, the requested changes were operated by the authors over a long period of time, which denotes seriousness and devotion to the subject addressed.

Reviewer 3 Report

No enough changes was made from the authors. So, I do not recommend this work for publication.

There is no deep discussion of the results.

No device discussion and characterization (I already mentioned in the last review).

I already said to the authors.

Reviewer 4 Report

Minor corrections of English language are required.